# Multifunctional Coding-Feeding Metasurface Based on Phase Manipulation

**DOI:** 10.3390/ma15197031

**Published:** 2022-10-10

**Authors:** Guo-Shuai Huang, Si-Jia Li, Zhuo-Yue Li, Xiao-Bin Liu, Cheng-Yuan He, Huan-Huan Yang, Xiang-Yu Cao

**Affiliations:** 1Information and Navigation College, Air Force Engineering University, Xi’an 710077, China; 2Xi’an Satellite Control Center, Xi’an 710043, China; 3State Key Laboratory of Millimeter Waves, Southeast University, Nanjing 210096, China; 4Shaanxi Key Laboratory of Artificially-Structured Functional Materials and Devices, Air Force Engineering University, Xi’an 710051, China

**Keywords:** coding-feeding metasurface, emission, scatting, OAM generation, RCS reduction

## Abstract

Multiple functionalities on a shared aperture are crucial for metasurfaces (MSs) in many applications. In this paper, we propose a coding-feeding metasurface (CFMS) with the multiple functions of high-gain radiation, orbital angular momentum (OAM) generation, and radar cross-section (RCS) reduction based on phase manipulation. The unit cell of the CFMS is composed of a rectangular emission patch and two quasi-Minkowski patches for reflective phase manipulation, which are on a shared aperture. The high-gain radiation and multiple modes of ±1, ±2, and ±3 OAM generation were realized by rationally setting the elements and the phase of their excitation. The CFMS presents a broadband RCS reduction of 8 dB from 3.18 GHz to 7.56 GHz for y-polarization and dual-band RCS reduction for x-polarization based on phase interference. To validate the concept of the CFMS, a prototype was fabricated and measured. The results of the measurement agree well with the simulation. A CFMS with the advantages of light weight and low profile has potential application in detection and wireless communication systems for stealth aircraft.

## 1. Introduction

Metasurfaces (MSs), as two-dimensional metamaterials, are artificial materials with periodic or aperiodic structures in sub-wavelength [1,2,3,4]. Due to the distinctive manipulation of electromagnetic (EM) waves, MSs attract great interest from researchers. Particularly, MSs can flexibly control the phase front over traditional methods through introducing discontinuities. In this regard, numerous devices and applications have been achieved, such as invisibility cloaking, perfect lensing, energy harvesters, polarization converters, and many others [5,6,7,8,9,10,11,12]. Furthermore, inspired by the binary concept, the groups of Engheta and Cui independently proposed the general notion of digital metamaterials, coding metamaterials, and programmable metamaterials, which developed a universal methodology to digitally control EM waves [13,14]. Generally, the coding metamaterials are composed of two kinds of unit cells, “0” and “1”, which have a phase difference of π. The EM waves can be manipulated by designing the sequences of coding elements “0” and “1” based on phase interference. As well, the concept of the coding metamaterials can be extended to 2-bit coding or higher. By loading the PIN diodes or other active devices into the metasurface, the reconfigurable coding metasurfaces lead to many exciting applications [15,16,17,18,19,20,21,22,23,24].

Recently, the scheme of a shared aperture has been momentous for multifunctional metasurfaces. For reflective metasurfaces, the multifunctionality of the early studies is usually limited in the same property. All the possible polarizations (linear, elliptic, and circular) or two-polarization conversions (from linearly-polarized wave to circularly-polarized wave, and from linearly-polarized wave to cross-linearly-polarized wave) have been realized through reflective metasurfaces, and the linearly polarized can be perfectly converted to the cross-polarized in certain conditions [25,26]. With the unit cells coupled by diodes, the linear polarization can be electrically controlled at the resonant frequency, or the elliptical polarization can be tuned off by the resonance [27]. Furthermore, by integrating the novel unit microstructure and optimized array, multiple functions have been realized on a shared aperture of the MS, including polarization conversion, RCS reduction, beam deflection, diffuse scattering, and vortex beam generation [28,29,30]. Moreover, the polarization, amplitude, and phase of EM waves can be manipulated by the transmissive multifunctional metasurface. Two functions of polarization conversion (linear to dual-circular polarization) and polarization selection is achieved with a composite transmission metasurface [31]. Based on the theory of coding metasurfaces, multiple functions of wave front shaping, pencil beam or multi-beam generation, and beam scanning have been realized through proper arrangement of the coding array [32,33].

Integrating reflection and transmission, hybrid metasurfaces realize various and numerous electromagnetic functions. For example, frequency-dependent functions can be realized by stacking different kinds of surfaces [34]. With chiral or anisotropic structures, polarization-dependent functions have been achieved in the unit aperture [35,36,37]. Recently, by introducing the coding concept, beam deflection, diffuse scattering, and vortex beam generation were realized by a transmission-reflection integrated coding metasurface [38]. Moreover, when loaded with electronically controlled PIN diodes, manipulation of the near/far-field patterns and the transmission/reflection modes for EM waves can be achieved in real time [39]. However, there are few reports about a metasurface with the functions of simultaneous radiation and reflection with a single layer.

In this work, we propose a coding-feeding metasurface (CFMS) to control reflective beams and emitting EM waves simultaneously. An optimized meta-atom consisted of a radiation patch and quasi-Minkowski micro-structure with a cell length of 0.27λ at 4 GHz was designed to achieve independent phase responses for different polarization incidences. The meta-atom and one with a rotation angle of 90 deg were employed as the “0” and “1” elements. With this checkerboard structure, the CFMS can reduce the RCS significantly. In addition, high-gain radiation and multimodal OAM generation were achieved by rationally setting the elements and their phase of excitation. The performance was numerically and experimentally demonstrated using radiation and scattering characteristics. The work may encourage potential applications in radar and wireless communication systems.

## 2. Designs and Methods

The multifunction diagram of the CFMS is shown in Figure 1. The unit cell was composed of a rectangular radiation patch and two quasi-Minkowski patches metal-printed onto a thin layer of substrate, which was backed by a metallic ground with a sub-miniature-a (SMA) port on the bottom. The unit cells and their characteristics are depicted in Figure 2, where the substrate was chosen as F4B with *ε**_r_* = 2.65 and tan *δ* = 0.0013. The patches and the ground were all copper, with a thickness of 0.036 mm and conductivity of 5.8 × 10^7^ S/m. The detailed geometry of unit 0 was as follows: *L* = 20 mm, *a*_1_ = 9.5 mm, *a*_2_ = 2.38 mm, *l_w_* = 9.5 mm, *lp* = 16.8 mm, *d* = 2.4 mm, *h* = 4 mm (for unit 1, *a*_1_ = 1.69 mm, *a*_2_ = 1.13 mm, *l_w_* = 8 mm, *lp* = 18.2 mm, *d* = 4.6 mm, the others were the same as unit 0).

For CFMSs, the reflection phase can be controlled by the size of the quasi-Minkowski micro-structures. The reflected phase changes more dramatically for smaller ones with y-polarized incidence. Therefore, we chose the proper size with a 180° ± 24° phase difference to manipulate the scattering based on the phase interference. In the design, we adopted a rectangular patch as the radiator. By optimizing the configuration and the feeding position, impedance matching could be achieved for the CFMS.

Due to the periodic boundary condition for the unit cell, we adopted 3 × 3 elements as a subarray to reduce differences caused by changes of boundary conditions between the unit cell and the CFMS array. The 3 × 3 element “0” was adopted as subarray “0” for the coding matrix, and that with the rotation angle of 90 deg was denoted as subarray “1”. The coding matrix of the CFMS is “0110/1001/1001/0110” in Figure 1b.

The commercial software CST Microwave Studio (version 2020, Computer Simulation Technology, Darmstadt, Germany) was employed to verify the properties of the unit cell and MS array. As shown in Figure 2b, the S_11_ of unit cells “0” and “1” was below −10 dB from 3.61 GHz to 4.36 GHz in radiation mode, which indicated impedance matching. Figure 2c–e shows the reflection amplitude and phase results of unit cells “0” and “1”. Because of the property of reciprocity, the reflection amplitude results for x-polarization were the same as the S_11_ data. It also shows the perfect reflection in the y-polarized incidence. The phase difference of about 180° ± 24° between unit 0 and unit 1 was from 4.56 GHz to 6.92 GHz for the *y*-polarized incidence and was in the band of 3.95–4.13 GHz and 11.02–11.50 GHz for the x-polarized incidence, which indicated that the unit cells of the CFMS can be used to manipulate the scattering.

With normally incident waves, the far-field scattering patterns of the CFMS can be expressed as:(1)$$E_{total}=E_{subarray}\cdot \sum_{m=1}^{M}\sum_{n=1}^{N}\exp\left\{-j\left\{kL\sin\theta \left[(m-\frac{1}{2})\cos\phi +(n-\frac{1}{2})\sin\phi \right]+\Theta (m,n)\right\}\right\}$$
where *E_subarray_* is the electric field intensity of the scattering for a subarray and *L* is the dimension of the subarray. *φ* and *θ* stand for the azimuth and elevation angles, respectively. Θ(*m*, *n*) is the phase difference of the scattering electric field between the adjacent subarrays.

The directivity function *D*(*φ*, *θ*) of the scattering field for the CFMS can be obtained as:(2)$$D(\varphi ,\theta )=\frac{4\pi {\left|E_{total}\right|}^{2}}{\int_{0}^{2\pi }\int_{0}^{2\pi }{\left|E_{total}\right|}^{2}\sin\theta d\phi d\theta }$$

From Equations (1) and (2), it is theoretically verified that the scattering beams and RCS of the CFMS can be significantly reduced based on the subarray and the coding matrix.

## 3. Results of Simulation and Measurement

Figure 3 shows the simulated scattering patterns at 4.79 GHz for a metal plate, the CFMS of all “0” units, and the CFMS checkerboard structure with the same area. The destructive interference has been chosen as the principle for arranging the coding elements. Based on the reflective phase difference of 180 deg between unit “0” and unit “1”, it is possible to control the scattering beams with the coding matrix. For example, when we adopted the same kind of elements for the CFMS, the normal scattering beam was strong and the scattering beam was the same as that of a metal plate, as given in Figure 3a,b. When the CFMS was arranged as a checkerboard structure, the normal scattering beam was separated into four side beams, as given in Figure 3c, which led to significant monostatic RCS reduction. Consequently, the proposed CFMS can effectively manipulate the scattering beam.

Figure 4 shows the simulated near electric fields of scattering for the metal plate and CFMS on the diagonal plane with incident *y*-polarized normal EM waves at 4.79 GHz and 6.71 GHz. As shown in Figure 4, it is obvious that the near electric intensity of the metal plate was much more than that of the CFMS in the normal direction. In contrast, the near electric fields of the normal reflection have almost vanished and have been redistributed to other directions by the proposed CFMS from Figure 4b,d. Thus, the near electric fields of the scattering have been controlled by the CFMS.

Based on theories of antenna arrays, the CFMS can achieve high-gain radiation when the SMA ports are fed, and impedance matching can be achieved by optimizing the configuration and feeding position. Figure 5 shows the characteristics of emission for the CFMS. From the radiation patterns with the unit of dBi in Figure 5a–c, it can be seen that the realized gain is 13.5 dBi at 3.98 GHz; the 3 dB beam width was 22.5 deg in both xoz-plane and yoz-plane at 3.98 GHz, and the side lobe was less than −15 dB. The front-to-back ratio of 33.9 dB was obtained for the CFMS at 3.98 GHz. What’s more, the components of cross-polarization at the direction of the main lobe were much less than those of co-polarization. Consequently, high-gain radiation can be realized by the CFMS.

For the CFMS with the phase-shifting feeding scheme (0°, 45°, 90°, 135°, 180°, 225°, 270°, 315°, clockwise 45-degree rotation), the OAM beam with −1 mode was generated in Figure 6a,b,e. The OAM beam with +1 mode was realized as the anticlockwise 45-degree rotation. The feeding position is dark red in Figure 6a. By adjusting the phase-shifting scheme (as 0°, 90°, 180°, 270°, 0°, 90°, 180°, 270° or 0°, 135°, 270°, 45°, 180°, 315°, 90°, 225°), the OAM beams with mode ±2 and mode ±3 could be realized by the CFMS. The 3D patterns, phases, and amplitude distributions of near electric fields for OAM generation with different modes are given in Figure 6b–f. From Figure 6e–g, we can see that the rotation direction of the vortex phase for OAM with +*l* (*l* = 1, 2, and 3) mode are contrary to that with −*l* (*l* = 1, 2, and 3) mode. Yet, the amplitude of the near-field distribution for OAM with +*l* mode is same as that with −*l* mode. It can be observed that the simulated results show a good performance for multimodal OAM generation for CFMS.

A prototype was fabricated and measured to validate the design and multiple functions of the CFMS. The CFMS prototype was fabricated using circuit board printing technology. The radiation and scattering characteristics were measured using an Agilent N5230C vector network analyzer. The function of scattering regulation could be reflected by the monostatic RCS reduction, which is depicted in Figure 7b. It is clear that significant monostatic RCS reduction can be achieved at multiple bands, which is more than 10 dB in the bands of 3.21–3.45 GHz, 4.05–4.58 GHz, 4.60–5.31 GHz, and 6.62–7.45 GHz in the y-polarized incidence. When the incident wave was x-polarized, the monostatic RCS reduction in the band of 4.02–4.25 GHz and 11.18–11.55 GHz was more than 10 dB. Moreover, the 8 dB RCS reduction can be realized from 3.18 GHz to 7.56 GHz for y-polarization. The y-polarized reduction peaks were 22.5 dB and 21.9 dB at 4.23 GHz and 6.88 GHz, respectively. This is approximately identical to the results of the simulation, which had resonances of 4.78 GHz and 6.65 GHz. The x-polarized peaks appeared at 4.19 GHz and 11.21 GHz, which was close to the simulation of 3.98 GHz and 11.03 GHz. The measured scattering patterns in diagonal sections were orthogonal to the xoy-planes of the CFMS illustrated in Figure 7c,d under y-polarized incidences. From the measured results at 4.79 GHz, the CFMS can significantly reduce the scattering and the reflective beams were almost uniformly distributed in four directions. The measured and simulated data were in good agreement.

The simulated and measured results verify the function of radiation for the CFMS in Figure 8. Figure 8a shows the S_11_ in the full-wave simulation and measurement respectively. From 3.69 GHz to 4.18 GHz, the magnitude of S_11_ was below −10 dB, which means good impedance matching and a potential emitting property. In Figure 8b,c, it can be seen that the gain was 13.25 dB and the half-beam width was 20.2 deg and 18.9 deg for the E-plane and H-plane, respectively. Moreover, the cross-polarized pattern was significantly less than the co-polarized one. Consequently, the CFMS exhibits excellent directivity and low cross-polarization, and the measurement agrees well with the simulation.

Figure 9 shows the experimental results of OAM generation for the CFMS. The near-field measurement system is depicted in Figure 9a, and the phase distribution of mode −2 is given in Figure 9c. The prototype of the CFMS is given in Figure 9b. The scanning area was 180 mm × 180 mm, with a step size of 3 mm. The phase distribution of OAM order −2 was clear, which verified the generation of OAM. The measured far-field radiation patterns of OAM patterns with −2 mode are given in Figure 9d. It can be seen that there are two nulls at the direction of propagation. The measured results are consistent with the simulated data. It is necessary to note that the difference between the simulation and the measurement can be attributed to the power divider, the fabricated accuracy of the prototype, the welds of the SMA ports, and the effect of the measurement environment.

Table 1 gives a comparison of scattering and radiation performance for this work with the other references. The metasurface in [28] can only obtain high-gain radiation, yet the scattering beam cannot be manipulated. Broadband RCS reduction and effective radiation can be realized by the metasurface antenna in [37]; however, its thickness of 6.508 mm is too thick, and it is very difficult to control the layer of air. From Table 1, it can be clearly seen that this thin CFMS not only achieves wide band RCS reduction and obvious RCS reduction peaks, but also realizes high-gain radiation.

## 4. Conclusions

To sum up, the coding-feeding metasurface was proposed, designed, and manufactured to realize its multifunctionality. The theory of interference and realization of phase manipulation was elucidated and theoretically analyzed. The multiple functions of high-gain radiation, RCS reduction, and OAM generation were verified by the results of both simulation and measurement. The concept of a CFMS integrates radiation and scattering manipulation creatively and achieves OAM generation with an ingenious coding arrangement. Consequently, the CFMS will be a good choice for the integrated control of radiation and scattering or other multifunctional applications.

## Figures and Tables

**Figure 1 materials-15-07031-f001:**
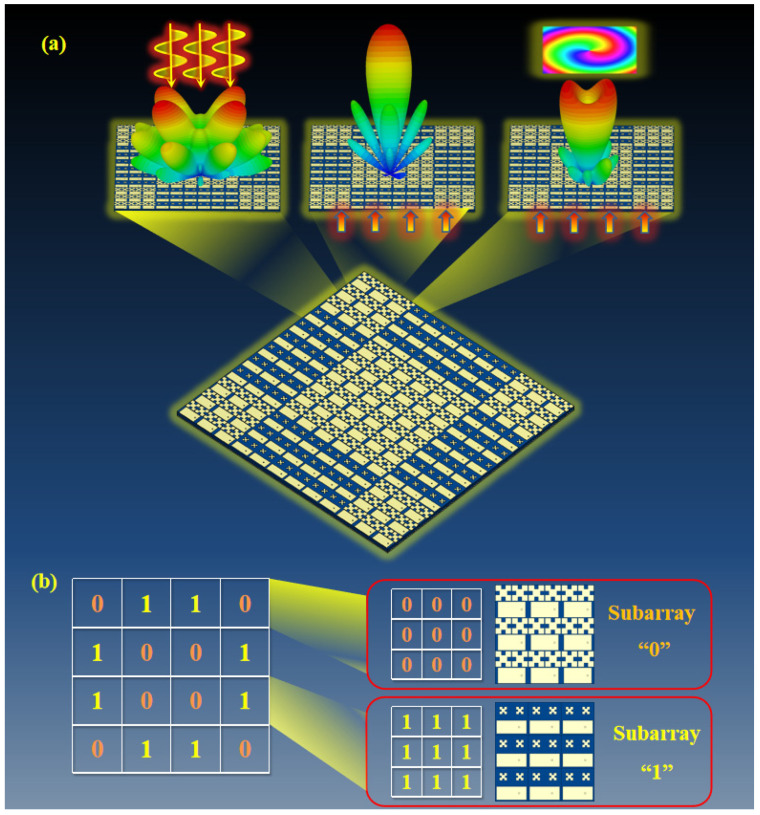
Schematic diagram of the CFMS with the multiple functions of RCS reduction, high-gain radiation, and OAM generation. (**a**) The functions of scattering beam modulation, high-gain radiation, and OAM generation. (**b**) Coding matrix of the CFMS based on subarray ”0” and subarray “1”. The F4B with *ε_r_* = 2.65 and *tanδ* = 0.0013 was chosen as the substrate. The rectangular radiation patch, two quasi-Minkowski patches, and ground were all copper, with a thickness of 0.036 mm and conductivity of 5.8 × 10^7^ S/m.

**Figure 2 materials-15-07031-f002:**
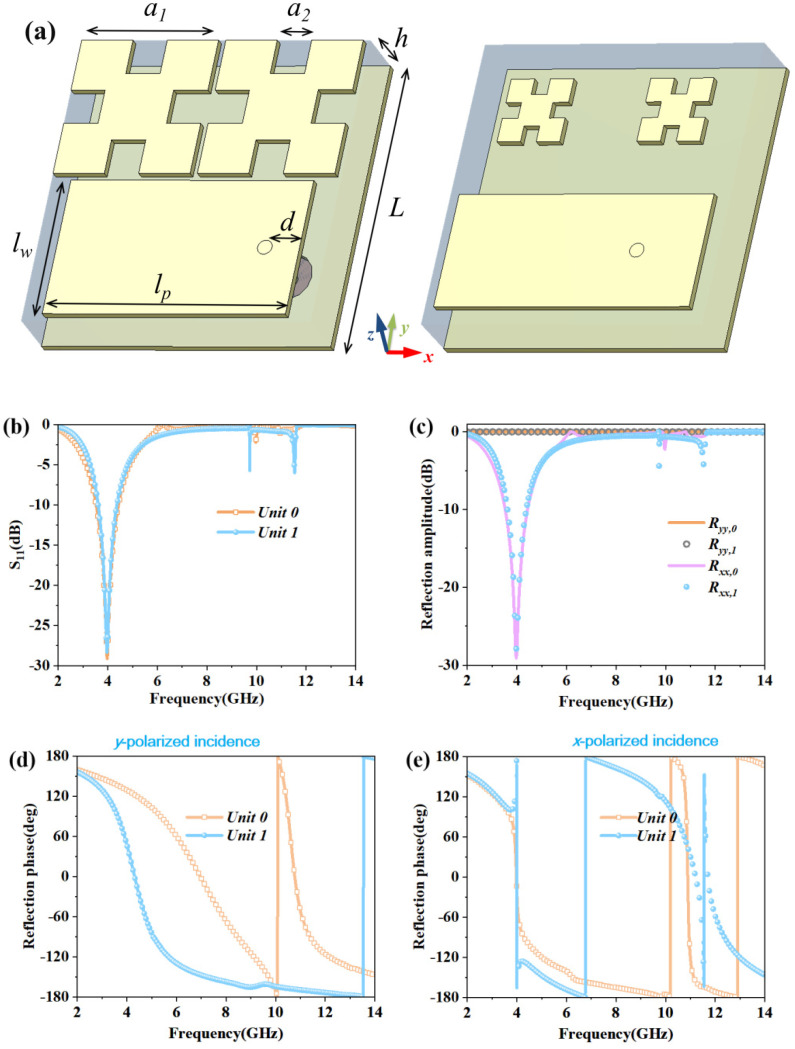
Geometry and characteristics of unit cells “0” and “1”. (**a**) The unit cells “0” and “1”; (**b**) Simulated amplitude results of reflection for unit cells “0” and “1” with radiation mode; (**c**) Reflection amplitude of unit cells “0” and “1” in plane incident waves; (**d**,**e**) Reflection phase of unit cells “0” and “1” for *y*- or *x*-polarized plane incident waves.

**Figure 3 materials-15-07031-f003:**
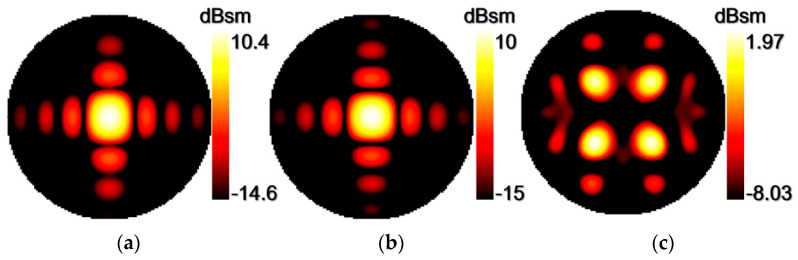
Simulated 3D scattering patterns for a metal plate, the CFMS of all “0” units and the CFMS with checkerboard structure with the same area at 4.79 GHz. (**a**) 3D scattering pattern of a metal plate. (**b**) 3D scattering pattern for the CFMS of all “0” units. (**c**) 3D scattering pattern of the CFMS with checkerboard structure.

**Figure 4 materials-15-07031-f004:**
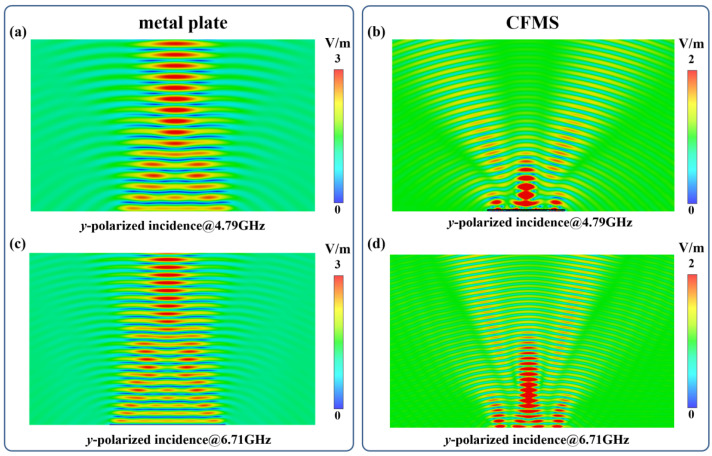
Simulated near electric fields of scattering for a metal plate and the CFMS with the same area on the diagonal plane with the *y*-polarized incidence. (**a**) Near electric fields of scattering for a metal brick at 4.79 GHz. (**b**) Near electric field of scattering for the CFMS at 4.79 GHz. (**c**) Near electric field of scattering for a metal brick at 6.71 GHz. (**d**) Near electric field of scattering for the CFMS at 6.71 GHz.

**Figure 5 materials-15-07031-f005:**
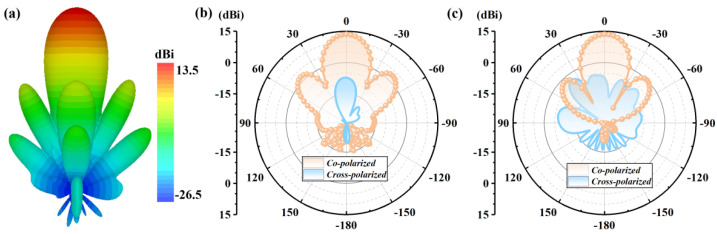
Simulated radiation patterns for the CFMS at 3.98 GHz. (**a**) 3D radiation patterns of the CFMS at 3.98 GHz. (**b**) Radiation patterns of co- and cross-polarized components in the xoz-plane for the CFMS. (**c**) Radiation patterns of co- and cross-polarized components in the yoz-plane for the CFMS.

**Figure 6 materials-15-07031-f006:**
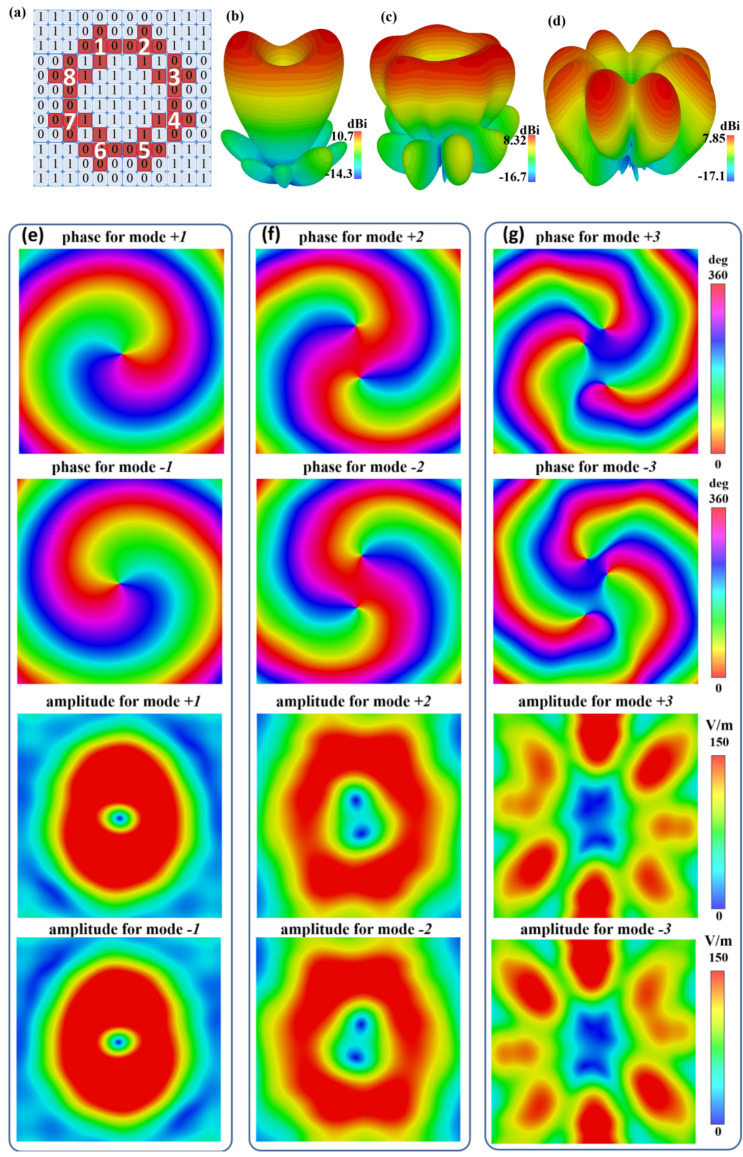
Diagram of the feeding positions and simulated results of OAM for the CFMS. (**a**) Diagram of the feeding positions. The eight subarrays were chosen to generate the OAM beam. (**b**–**d**) 3D radiation patterns of the OAM with 1, 2, 3 modes. (**e**–**g**) Phase and magnitude results of near electric field distributions on plane for the CFMS with the OAM in different modes. The OAM beams in +1, +2, and +3 mode can be respectively realized as the incident phases in an anticlockwise feeding phase with 45-, 90-, and 135-degree rotation. Correspondingly, the OAM beams in −1, −2, and −3 modes were achieved as the incident phases with clockwise 45-, 90-, and 135-degree rotation.

**Figure 7 materials-15-07031-f007:**
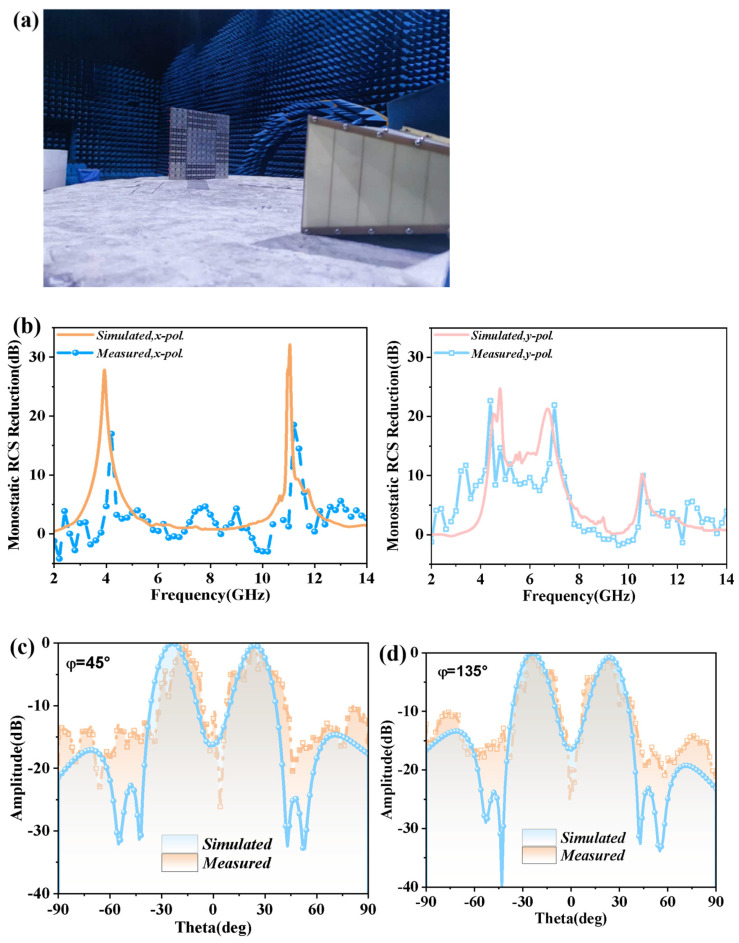
Environment of the measurement and measured scattering results. (**a**) Environment of the measurement for scattering. (**b**) Measured and simulated monostatic RCS reduction with *x*- or *y*-polarized incidences. Measured and simulated scattering patterns of (**c**) φ = 45° plane and (**d**) φ = 135° plane for the CFMS with the y-polarized incidence at 4.79 GHz.

**Figure 8 materials-15-07031-f008:**
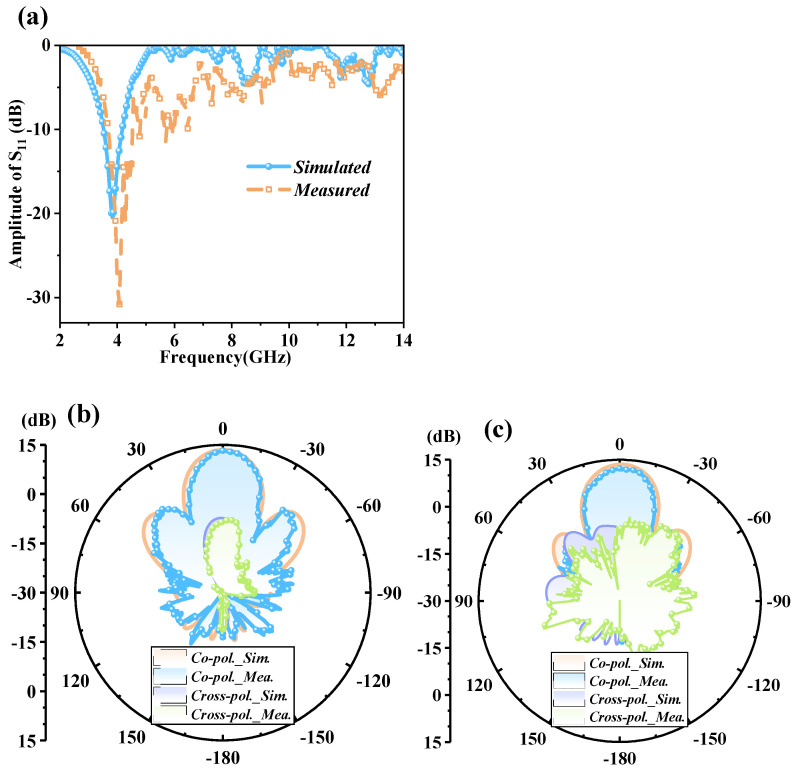
Measured and simulated amplitudes of S_11_ and radiation patterns at 4 GHz for the CFMS. (**a**) Measured and simulated amplitudes of S_11_ for the CFMS. (**b**) Co- and cross-polarized radiation patterns on the E-plane. (**c**) Co- and cross-polarized radiation patterns on the H-plane.

**Figure 9 materials-15-07031-f009:**
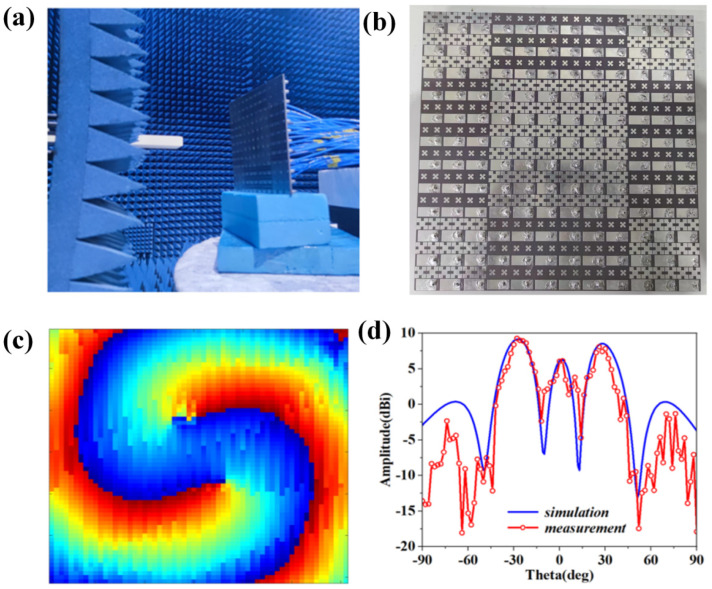
Environment of the measurement for near-field and measured results. (**a**) Environment of the measurement for the CFMS. (**b**) Prototype of the CFMS. (**c**) Measured phase distribution of OAM with −2 mode on the plane. (**d**) OAM radiation patterns in xoz-plane.

**Table 1 materials-15-07031-t001:** Comparison of scattering and radiation performance for this work with other references.

Ref.	RCS Reduction Band	RCS Reduction Peak	Radiation	Gain	Thickness
[28]	none	none	Ok	26 dBi	4.0 mm
[29]	6.7–24 GHz	16 dB	Ok	<5 dBi	3.0 mm
[36]	3.0–4.3 GHz	16.8 dB	Ok	7.0 dBi	6.0 mm
[37]	5.0–18 GHz	30 dB	Ok	9.5 dBi	6.508 mm
This work	3.18–7.56 GHz	22.5 dB	Ok	13.5 dBi	4.0 mm

## Data Availability

Not applicable.

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
