# Peer review of "Multifunctional Coding-Feeding Metasurface Based on Phase Manipulation"

_materials, 2022, doi:10.3390/ma15197031_

Round 1

Reviewer 1 Report

Huang et.al. have designed and validated multifunctional coding-Feeding metasurface based on phase manipulation. The manuscript has been written nicely and data are presented with clarity. The results of the measurement are interesting and find potential application in communication system. I have some minor comments and they are highlighted below.

1.     The perspective view of unit “0” and “1” should be clearer. At present it doesn’t give over all view of the unit.

2.     The fabricated Code-Feeding metasurface planar structure should be shown in the figure.

3.     In figure 7, the agreement between experimental and simulated data hasn’t been great. The author should explain this.

4.     In figure 9b, the color scale is missing and should be corrected.

5.     The polar plot shown in the figure 5b and 5c doesn’t signify the magnitude of the radiation pattern in E- AND H-plane of the CFMS. Only it gives directional emission and missing other information. And hence it should be updated.

Author Response

The details are given in the response letter

Reviewer 2 Report

The authors have presented a "Multifunctional Coding-Feeding Metasurface Based on Phase Manipulation" which aims to develop a coding-feeding metasurface (CFMS) with multiple functions of high gain radiation, orbital angular momentum (OAM) generation and radar cross section (RCS) reduction based on phase manipulation. The unit cell of CFMS is composed of a rectangle emission patch and two quasi-Minkowski patches for reflective phase manipulation, which are on a shared aperture. The high gain radiation and multimodal OAM generation are realized by rationally setting the elements and their phase of the excitation. Besides, CFMS presents the broad-band RCS reduction based on phase interference. To validate the concept of CFMS, a prototype is fabricated and measured. The results of measurement agree well with the simulation. CFMS with advantages of lightweight and low profile has potential application in the detection system and wireless communication system for stealth aircraft. The proposed manuscript is interesting for readers of the journal and can be considered for publication after the authors resolve the following issues:

1.) The abstract is written nicely. But it consists of too much theoretical discussion and it requires numerical results to support the claim and make it more impactful.

2.) The latest literature review is missing from the manuscript. The authors can refer to the following latest advances in encoding and detection systems developed with the help of metasurfaces and phase change materials:

Detection of cancer with graphene metasurface-based highly efficient sensors, Diamond and Related Materials, Volume 129, 2022, 109367, ISSN 0925-9635, https://doi.org/10.1016/j.diamond.2022.109367 (https://www.sciencedirect.com/science/article/pii/S0925963522005490)

3.) The authors have mentioned the 90-degree rotation but they have forgotten to include the direction of rotation. i.e., clockwise or anticlockwise?

4.) There are so many typos as well as grammatical mistakes in the manuscript. i.e. page 5 line no. 117, below is misspelled, line no. 122 and so on. I suggest authors revise the whole manuscript for such mistakes.

5.) The materials employed for the pattern and substrate should be clearly mentioned in the text as well as in Fig. 1 itself.

6.) The discussion is comparatively less. I suggest authors discuss Fig. 3 to 6 in detail.

7.) Fig. 7 (b) can be divided into two parts as the results are not clearly visible. I suggest authors split the Fig. into two parts. i.e, x-polarized and y-polarized with simulation and experimental results.

8.) Fig. 8(b, c) are not visible. The quality of fig. should be enhanced as we can't clearly visualize the labels and what they want to convey. The authors should revise this figure.

9.) A comparison study should be presented at the end of section 3 to demonstrate the high performance of the proposed work and the significance of the study should also be clearly stated in the manuscript.

The manuscript in its current form can't be accepted. I suggest authors carefully revise the manuscript keeping in mind the abovementioned points and resubmit it.

Author Response

The details are given in the response letter.

Round 2

Reviewer 2 Report

The authors have incorporated all the suggestions and improved the manuscript. I suggest that the manuscript can now be published in its current form.